# Hysteresis in the Thermo-Responsive Assembly of Hexa(ethylene glycol) Derivative-Modified Gold Nanodiscs as an Effect of Shape

**DOI:** 10.3390/nano12091421

**Published:** 2022-04-21

**Authors:** Joshua Chidiebere Mba, Hideyuki Mitomo, Yusuke Yonamine, Guoqing Wang, Yasutaka Matsuo, Kuniharu Ijiro

**Affiliations:** 1Graduate School of Life Science, Hokkaido University, Kita 10, Nishi 8, Kita-Ku, Sapporo 060-0810, Hokkaido, Japan; joshuachidiebere.mba.x6@elms.hokudai.ac.jp; 2Research Institute for Electronic Science, Hokkaido University, Kita 21, Nishi 10, Kita-Ku, Sapporo 001-0021, Hokkaido, Japan; yonamine@es.hokudai.ac.jp (Y.Y.); matsuo@es.hokudai.ac.jp (Y.M.); 3College of Food Science and Engineering, Ocean University of China, 5 Yushan Road, Qingdao 266003, China; gqwang@ouc.edu.cn

**Keywords:** gold nanodiscs, self-assembly, thermo-responsive, shape effects, hysteresis, thermo-dynamic, active plasmonics

## Abstract

Anisotropic gold nanodiscs (AuNDs) possess unique properties, such as large flat surfaces and dipolar plasmon modes, which are ideal constituents for the fabrication of plasmonic assemblies for novel and emergent functions. In this report, we present the thermo-responsive assembly and thermo-dynamic behavior of AuNDs functionalized with methyl-hexa(ethylene glycol) undecane-thiol as a thermo-responsive ligand. Upon heating, the temperature stimulus caused a blue shift of the plasmon peak to form a face-to-face assembly of AuNDs due to the strong hydrophobic and van der Waals interactions between their large flat surfaces. Importantly, AuNDs allowed for the incorporation of the carboxylic acid-terminated ligand while maintaining their thermo-responsive assembly ability. With regard to their reversible assembly/disassembly behavior in the thermal cycling process, significant rate-independent hysteresis, which is related to their thermo-dynamics, was observed and was shown to be dependent on the carboxylic acid content of the surface ligands. As AuNDs have not only unique plasmonic properties but also high potential for attachment due to the fact of their flat surfaces, this study paves the way for the exploitation of AuNDs in the development of novel functional materials with a wide range of applications.

## 1. Introduction

Recent advances in the fields of nanoscience and nanotechnology have seen the development of a plethora of nanoparticle types including metallic nanoparticles [1,2,3,4]. Among them, gold nanoparticles (AuNPs) have been widely exploited across diverse applications due to the fact of their unique surface plasmon resonance [5], biocompatibility [6], and ease of functionalization by surface modification using amino, thiol, or carboxyl residues on small molecules [7,8] or polymers [9]. The shape as well as the size of AuNPs are essential for the control of their optical, electronic, and physical properties [10]. Further, assembly of AuNPs promises the seamless design and fabrication of nanodevices with novel physicochemical properties for diverse uses.

Self-assembly, which refers to the association of individual components of a material into well-ordered patterns, often generates materials with emergent features that can vary from the properties of the components. It is one of the practical strategies for making ensembles of nanostructures with exceptional properties and functionalities [11]. When functionalized with stimulus-responsive ligands, AuNPs alter their physical and/or chemical properties at their surfaces upon encountering external stimuli, such as solution pH [12,13], temperature [14,15,16], and light [17,18], and form assembled structures. The ability to reversibly control nanoparticle assemblies with external stimuli accelerates tunable plasmon coupling for innovative applications [19]. Thus, in recent years, the assembly of AuNPs into well-defined two- and three-dimensional nanostructures has attracted a great deal of attention [20,21]. To date, there have been numerous reports on the utilization of AuNPs self-assembly as probes [22,23,24,25], sensors [26], and in electronics [27,28,29] as well as for biomedical purposes such as in drug delivery [30], cancer detection [31,32], and cancer photothermal therapy [33,34,35]. From the above, it can be seen that gold nanospheres (AuNSs) and gold nanorods (AuNRs) are predominantly used for studies on thermo-responsive assembly, even though there are many other differently shaped nanoparticles that show valuable plasmonic properties.

Gold nanodiscs (AuNDs), which are plasmonic nanostructures with two circular-shaped flat surfaces, are another type of attractive anisotropic nanoparticles. Like other anisotropic AuNPs, AuNDs possess unique properties including a wide range of resonant wavelengths in the visible and NIR regions, tunable ratio of light absorption to scattering, and two large flat surfaces with circular symmetry [36]. Due to the fact of their flat surfaces, AuNDs can assemble either on the disc plane, providing configurational symmetry breaking on plasmon coupling, or assemble along the direction perpendicular to the disc plane for use as planar hot spots for surface-enhanced Raman scattering (SERS). In line with these unique properties, stimuli-responsive assemblies of AuNDs have the capability to provide enhanced or emergent functions. Yet, to date, AuNDs have been under-represented in terms of studies in comparison with other AuNP shapes.

Our group has previously investigated the thermo-responsive assembly of AuNPs (AuNSs and AuNRs) modified with a self-assembled monolayer (SAM) of hexa(ethylene glycol) (HEG) derivatives with ethyl (**C2**) or methyl (**C1**) heads and alkyl-thiol tails [15,37,38]. The thermo-responsive assembly of the AuNPs resulted from hydrophobic interactions between AuNPs due to the dehydration of the HEG part of the ligands upon heating. Compared to other reports using thermo-responsive polymers [39], our surface ligands, as small molecules, show significant shape-dependent effects, in particular surface curvatures, of particles in terms of their thermo-responsiveness (assembly temperatures). By utilizing this curvature dependence of our HEG-SAM-based modification, we reported the reversible hierarchical assembly of AuNRs via a simple surface modification with a single kind of ligand as a new strategy for the design and fabrication of stimuli-responsive sophisticated assemblies [15]. This suggests that we can draw out further hidden potentials from various anisotropic-shaped nanoparticles on assembly. At present, however, no similar studies have been conducted using AuNDs, despite their unique properties and prospective applications. Thus, in this study, we synthesized AuNDs, modified them with HEG derivatives to provide thermo-responsiveness, and then investigated their thermo-responsive assembly/disassembly behaviors (Figure 1), demonstrating unique shape-related phenomena as a higher degree of hysteresis on reversible assembly/disassembly.

## 2. Materials and Methods

### 2.1. Materials

Tetrachloroauric acid (HAuCl_4_·3H_2_O), sodium borohydride, sodium iodide, tris(2-carboxyethyl) phosphine hydrochloride (TCEP), and ascorbic acid were purchased from Merck (Sigma Aldrich Chemie GmbH, München, Germany). Sodium hydroxide was purchased from FUJIFILM Wako Pure Chemical Corp. (Osaka, Japan). Hexadecyl trimethylammonium bromide (CTAB) was purchased from Tokyo Chemical Industry Co., Ltd. (Tokyo, Japan). Citrate-stabilized gold nanospheres (40 nm) in aqueous solution were purchased from BBI Solutions (UK). HEPES was purchased from Dojindo Laboratories Co., Ltd. (Kumamoto, Japan). Ultrapure water was used to prepare all solutions for the experiments (Milli-Q Reference system). The thermo-responsive ligand with a methyl-terminated head referred to as **C1** was synthesized from [11-(methylcarbonylthio) undecyl] hexa(ethylene glycol) methyl ether (Sigma–Aldrich) according to previous reports [40,41]. The carboxylic acid-terminated ligand, 20-(11-mercaptoundecanyloxy)-3,6,9,12,15,18-hexaoxaeicosanoic acid (**COOH**), was purchased from Dojindo Laboratories Co., Ltd. (Japan). All commercially available reagents were used without further purification.

### 2.2. Synthesis of Gold Nanotriangles (AuNTs)

AuNTs were synthesized according to a previous report [42]. Briefly, 0.5 mL of 20 mM HAuCl_4_ solution was added to 36.5 mL of water. Next, 1 mL of a 10 mM aqueous solution of sodium citrate and 1 mL of 100 mM aqueous NaBH_4_ (ice-cold) solution were added with vigorous stirring for 2 min and kept over 4 h at 30 °C in an incubator to produce gold seeds. To prepare the triangular nanoplates, growth solutions were prepared as follows. A mixture of 108 mL of 50 mM CTAB solution and 54 μL of 0.1 M NaI solution was divided into three containers labeled with 1, 2, and 3. Containers 1 and 2 held 9 mL of the mixture and container 3 held the rest of the solution (90 mL). Next, 125 μL of 20 mM HAuCl_4_ solution, 50 μL of 100 mM NaOH, and 50 μL of 100 mM ascorbic acid were added to container 1 and to container 2. Next, 1.25 mL of 20 mM HAuCl_4_, 0.5 mL of 100 mM NaOH, and 0.5 mL of 100 mM ascorbic acid were added to container 3. Subsequently, 1 mL of the seed solution was added to container 1 with mild shaking. Then, 1 mL of container 1 solution was added to container 2. After approximately 5 s of shaking, the entire solution from container 2 was added to container 3 and kept at 30 °C. The supernatant was discarded after 24 hrs and the resultant AuNTs (greenish in color with strong absorption in the NIR) were redispersed in 20 mL of 25 mM CTAB.

### 2.3. Synthesis of Gold Nanodiscs (AuNDs)

Gold nanodiscs were synthesized from AuNTs via a comproportionation reaction according to a previous report with a few modifications [43]. Briefly, 20 μL of 10 mM HAuCl_4_ was added to 5 mL of the as-prepared AuNTs while vigorously mixing by vortex for 90 s. The solution was left overnight at 30 °C for 13 h. The AuNDs solution was purified by 2 cycles of centrifugation (9400× *g*, 8 min, 30 °C) to end the reaction and resuspended with 5 mL of 25 mM CTAB. To synthesize AuNDs of smaller diameter, a second comproportionation reaction was performed as above to further etch the synthesized AuNDs to a smaller diameter.

### 2.4. Surface Modification of AuNDs

The as-prepared AuNDs (1 mL) were purified twice by centrifugation (9400× *g*, 8 min, 30 °C), and resuspended in 1 mM CTAB solution. Before use, thiol ligands (**C1**) were mixed with tris(2-carboxyethyl) phosphine hydrochloride (TCEP) as a reductant for over 1 h at 30 °C. Solutions of mixed ligands were then prepared as follows: **C1:COOH** = 100:0, 99:1, 97:3, 95:5, and 90:10 (5 mM final concentration). Next, 100 μL of 2 mM CTAB was added to 100 μL of the mixed ligand and that solution was then added to 800 μL of AuNDs and incubated at 30 °C for 15 h. A second surface coating was performed for 24 h to ensure apposite surface modification. The surface-modified AuNDs were washed by three cycles of centrifugation with 10 mM HEPES buffer (pH 8.0) and 10 mM NaOH to remove free thiol ligands. The samples were finally resuspended in a 10 mM HEPES buffer (pH 8.0) for subsequent experiments.

### 2.5. Surface Modification of Gold Nanospheres (AuNSs)

Prior to surface modification, the surface coatings of 40 nm citrate-stabilized AuNSs were changed to CTAB micelles according to a previous report with some modifications [44]. This was to ensure the AuNSs had the same stabilizing substance and a positive surface charge like the CTAB-stabilized AuNDs. Briefly, 10 mL of the citrate-AuNSs was mixed with 10 mL of 10 mM CTAB and incubated for 24 h at 30 °C. The nanoparticles were purified by centrifugation (12,000× *g*, 15 min, 30 °C) and the surface coating with CTAB micelles was repeated as above to ensure sufficient replacement of the weakly bound citrate anions. Next, the 40 nm AuNSs were modified with a mixed solution of **C1** and **COOH** ligands (**C1:COOH** = 100:0 and 99:1) via a ligand exchange reaction as with the AuNDs. Briefly, 1 mL of CTAB-stabilized AuNSs was concentrated by centrifugation (12,000× *g*, 15 min) and 900 μL of the supernatant was removed. The concentrated AuNSs were then mixed with 100 μL solution of the mixed ligands followed by the addition of ultrapure water up to 500 μL and incubation for 24 hrs twice at 30 °C for a total of 48 h. The modified AuNSs were washed 3 times to remove the free thiol ligands by centrifugation (12,000× *g*, 15 min) and resuspended in 10 mM HEPES buffer (pH 8.0).

### 2.6. UV−Vis-NIR Spectroscopy

The UV−Vis-NIR spectra of the synthesized nanoparticles were measured using UV−Vis-NIR spectrophotometers (V-730 or V-770) with a PAC-743R Automatic 6 position Peltier cell changer (JASCO Corp., Tokyo, Japan). A heating rate of 1 °C/min and a waiting time of 5 or 30 min were applied for the measurement of the thermo-responsive assembly and disassembly. We defined the assembly (*T*_A_) and disassembly (*T*_D_) temperatures as that corresponding to the midpoint of the extinction change during the heating and cooling processes, respectively. The degree of hysteresis was determined as the difference between the assembly and disassembly temperature.

### 2.7. Dynamic Light Scattering (DLS) and Zeta Potential Measurement

The diameter and the zeta potentials of the nanoparticles were measured using a ZetaSizer Nano ZS (Malvern Panalytical Ltd., Morvin, UK).

### 2.8. Scanning Transmission Electron Microscope (STEM) Observation

STEM images were obtained using a HD-2000 (Hitachi High-Tech, Tokyo, Japan) with a 200 kV accelerating voltage. Samples were prepared by dropping 5 μL of the sample solution onto the collodion membrane-attached mesh (Nisshin EM, Tokyo, Japan). This was dried in a desiccator overnight before the images were taken. The obtained STEM images were used to determine the shape and diameter of the AuNDs. Over 200 discs were counted using the ImageJ software.

### 2.9. Scanning Electron Microscope (SEM) Observation

SEM images were obtained using an Ultimate Field Emission Scanning Electron SU-8230 (Hitachi High-Tech, Tokyo, Japan) to determine the shape and thickness of the AuNDs. When unable to verify the thickness of the AuNDs from the well-dispersed STEM images, we confirmed the thickness using a SEM, which is better equipped to give the desired result via tilting. Here, 60- and 105-AuND (97:3) samples were prepared by dropping 15 μL of the sample solution onto a silicon substrate. This was left to dry overnight before the images were taken with a working distance (WD) of 3.0 mm and 0 tilt.

### 2.10. Finite-Difference Time-Domain (FDTD) Simulation

The absorption spectra of the AuNDs in water were calculated using the FDTD (FullWAVE, Rsoft, Ossining, NY, USA) methods. The dielectric functions of Au and water were taken from previous reports [45,46]. The electromagnetic field propagated along the *z*-axis and oscillated along the *x*-axis. The incident electric field amplitude was normalized to 1.

## 3. Results and Discussion

### 3.1. Preparation of Thermo-Responsive AuNDs

AuNDs were synthesized from AuNTs via a comproportionation reaction, which is a type of chemical reaction that allows surface gold atoms to be oxidized in a self-limiting and tip-selective manner by the addition of HAuCl_4_ solution (Appendix A) [43]. First, AuNTs were synthesized according to a previous report [42]. The extinction spectrum of the synthesized AuNTs shows a localized surface plasmon resonance (LSPR) peak at approximately 1250 nm in the NIR region (Appendix A). The STEM images show they formed thin equilateral triangular shapes based on their high electron transmittance (Appendix A). The average edge length of one side of the AuNTs was determined from STEM images to be 163 ± 19 nm (Appendix A). After the comproportionation reaction was performed with the AuNTs, the LSPR peak showed a blue shift to 827 nm, indicating their shape changes to AuNDs (Figure 1a(blue)). We determined the diameter and the shape of the AuNDs by DLS analysis and STEM. The results of the DLS measurements revealed two peaks at approximately 10 and 100 nm (Figure 1b(blue)). These corresponded to the rotational and translational diffusion modes, respectively, which are peculiar to anisotropic nanoparticles, supporting the fact that we had successfully produced anisotropic disc structures [47,48,49,50]. STEM images provided the diameter of AuNDs as 105 ± 13 nm, hereafter designated as 105-AuNDs, indicating a narrow size distribution among the synthesized AuNDs (Figure 1c). To synthesize AuNDs of smaller diameter, the above synthesized AuNDs were resuspended in 25 mM CTAB and a second comproportionation reaction was performed. The shift to the shorter wavelength of 715 nm shown by the synthesized AuNDs after the second etching signifies a decrease in their aspect ratio (Figure 1a(red)). The decrease in the size of the AuNDs was revealed by the DLS results, as shown by the differences in their peaks at approximately 100 nm, and by STEM images showing them to be 60.4 ± 8.2 nm in diameter, hereafter designated as 60-AuNDs (Figure 1b(red),d). Scanning electron microscopy (SEM) images showed stacked structures of AuNDs that formed on the silicon substrate during the drying process, revealing the thicknesses of the 105-AuNDs and 60-AuNDs to be approximately 6–7 nm (Figure 1e,f).

Next, AuNDs, which were stabilized with CTAB as prepared, were modified with the thermo-responsive ligand (**C1**) mixed with different percentages (0, 1, 3, 5, and 10 mol%) of carboxylic acid-terminated ligand (**COOH**). The AuNDs modified with the mixture of **C1** and **COOH** ligands were designated as AuND-(XX:YY) in which XX represents the ligand ratio of **C1** and YY represents that of **COOH**. The zeta potentials of the CTAB-stabilized and ligand-modified AuNDs were measured in 10 mM HEPES buffer (pH 8.0) (Figure 1g,h). The zeta potential values changed from a positive charge of 51 mV for the CTAB-stabilized 105-AuNDs to a negative charge of −10 mV for **C1**-coated 105-AuNDs, designated as 105-AuND-(100:0). The others, 105-AuND-(99:1), -(97:3), -(95:5), and -(90:10), showed charges of −14, −20, −23, and −29 mV, respectively. The zeta potential values for 60-AuND-CTAB and 60-AuND-(100:0), -(99:1), -(97:3), -(95:5), and -(90:10) were 55, −12, −15, −19, −24, and −28 mV, respectively. The **COOH** content and zeta potential showed a good correlation (Appendix A). The results of the spectral analysis, DLS measurements, STEM, SEM, and zeta potential analysis support the successful synthesis and surface modification of the AuNDs.

### 3.2. Thermo-Responsive Assembly of AuNDs

The thermo-responsive assembly of AuNDs modified with the thermo-responsive ligand was investigated by heating the samples from 25 to 70 °C at a rate of 1 °C/min. Spectral changes were measured every 5 °C after 5 min of waiting time. The extinction spectra for 105-AuND-(100:0) showed a total decrease in not only the LSPR peak at approximately 850 nm but also absorption and scattering by AuNPs at shorter wavelengths below 500 nm during heating between 30 and 40 °C, suggesting precipitation of AuNDs due to the formation of assemblies in larger sizes (Appendix A). This thermo-responsive assembly of 105-AuND-(100:0) is thought to be driven by the dehydration of the ethylene glycol part of the **C1** ligand due to the increase in temperature [37]. The temperature for this AuND assembly (between 30 and 40 °C) is much lower than that of AuNSs (73 °C for 10 nm, 63 °C for 15 nm, and 53 °C for 20 nm in a diameter) [37] or AuNRs (approximately 65 °C for 33 × 14 nm) [15] but similar to that for silver nanocubes (AgNCs) of 25 nm in size [51]. This is in line with the curvature dependence of the assembly temperature (*T*_A_) in which the lower curvature at their flat surfaces leads to a lower *T*_A_. To examine the reversibility of the thermo-responsiveness of 105-AuND-(100:0), we cooled the solution temperature down to 25 °C. However, their spectra did not recover, indicating 105-AuND-(100:0) remained assembled (Appendix A). This result is inconsistent with those of our previous reports, where AuNRs and AuNSs modified with **C1** ligand (100%) showed reversible assembly/disassembly [15,37]. The irreversibility of 105-AuND-(100:0) was likely due to the flat surfaces of the AuNDs facilitating strong attraction against disassembly. While spheres attach at a point (zero dimension; 0D) and rods can attach along a line in a side-by-side (1D) structure, discs can attach across a plane in a face-to-face (2D) structure. This non-reversibility of the assembled 105-AuND-(100:0) is thus attributed to the strong hydrophobic and van der Waals interactions between the flat surfaces of 105-AuND-(100:0).

We hypothesized that an electrostatic repulsive force helps to induce disassembly of the strongly assembled AuNDs. Thus, AuNDs were modified with a **C1:COOH** ligand mixture of 97:3, shown as 105-AuND-(97:3). The spectral changes showed a decrease in the intensity of the plasmon peak at approximately 830 nm and the appearance of a new plasmon peak at approximately 700 nm upon heating to over 60 °C, indicating assembly formation in response to temperature (Figure 2a). When cooled, the extinction peaks returned to the original values at 45 °C (Figure 2b). This means 105-AuND-(97:3) disassembled and redispersed. Interestingly, their plasmon intensity changes and peak shift upon temperature change demonstrated a large hysteresis, which is a large difference between the *T*_A_ and disassembly temperature (*T*_D_) (Figure 2c, Table 1). This hysteresis is further discussed below. Thermal cycling experiments support the notion that the thermo-responsive assembly/disassembly is repeatable (Figure 2d). To clarify the assembly structures of the AuNDs, STEM imaging was performed. The 105-AuND-(97:3) sample was dried overnight on the TEM grid at 25 and 60 °C, below and around their assembly temperatures, respectively. The SEM image dried at 25 °C shows well-packed AuNDs attached on the grid in a flat surface, and the well-contrasted TEM image suggests single layered attachment (Appendix A). On the other hand, the SEM image dried at 60 °C shows disturbed attachment, and the TEM mode provides an unclear dark image probably due to the low transmittance from the stacked structures (Appendix A). Importantly, we could find face-to-face assemblies of the AuNDs in pancake-like nanostructures (Figure 2e). To clarify their assembly structures, we also performed FDTD simulation. Simulated spectra showed that face-to-face assembly caused a blue shift of the plasmon peak for AuNDs (Figure 2f). In more detail, FDTD simulation supports the notion that the decrease in peak intensity at approximately 830 nm, as shown in Figure 2a, represents a smaller number of dispersed AuNDs in solution, and the newly appeared peak at approximately 700 nm results from assemblies composed of two or three nanoplates. In other words, the FDTD simulation results support face-to-face assembly formation in solution.

To investigate the mechanism underlying their face-to-face assembly, we studied the effect of salt concentrations on 105-AuND-(97:3) assembly at 25 °C (Appendix A). As salt reduces electrostatic repulsions, an increase in salt concentration should induce assembly formation. The results revealed that a low salt concentration (10 mM) resulted in a small decrease in the peak intensity for 105-AuND-(97:3), and medium salt concentrations (20 and 30 mM) provided blue shifts in the plasmon peaks. These results are similar to those for the thermo-responsive assembly of 105-AuND-(97:3) in Figure 2a. This indicates that AuNDs can preferably form face-to-face assemblies. This was also observed in the SEM images in Figure 1e,f. On the other hand, the highly reduced electrostatic repulsion using 50 mM NaCl caused a red shift and broadening of the plasmonic peak, indicating the random aggregation or assembly of the stacked AuNDs. Although AuNDs preferably assemble across their flat surfaces due to the fact of their shape, these results suggest that shape itself is not a completely dominant factor in the formation of face-to-face assemblies. In essence, the curvature dependence on the thermo-responsive properties obtained from our thermo-responsive ligand (**C1**) should be the dominant factor leading to this face-to-face assembly [15,37,38].

### 3.3. Effects of Size and Shape on Thermo-Responsive Assembly

The size and shape of gold nanoparticles are key determinants of their interparticle interactions and are also critical in defining their properties. In particular, for AuNDs, their flat surfaces have the potential to provide strong particle–particle interactions as mentioned above. Thus, we investigated the effect of size on the thermo-responsive behavior using AuNDs with a smaller diameter, 60-AuNDs. The 60-AuND-(100:0) consistently showed a decrease in peak intensity and a blue shift of the plasmon peak upon heating (Appendix A). When cooled, the spectra did not return to the original values, suggesting the nanoparticles remained assembled as observed for 105-AuND-(100:0) (Appendix A). On the other hand, the 60-AuND-(97:3) showed reversible thermo-responsive assembly and disassembly (Figure 3a,b). Their plasmon intensity changes on changes in temperature revealed a large hysteresis (Figure 3c, Table 1). Thermal cycling experiments support the notion that the thermo-responsive assembly/disassembly was repeatable (Figure 3d). These results are the same as those for 105-AuNDs, even though their area of the flat surface was only one-third.

To further confirm the role of the shape of AuNDs in their unique thermo-responsive behaviors, the thermo-responsive assembly of AuNSs with a similar volume was investigated for comparison. As the volume of 60- and 105-AuNDs were calculated to be 20,000 and 52,000 nm^3^, respectively, AuNSs of 40 nm in diameter (40-AuNSs), which had a volume of 33,000 nm^3^, were used. To ensure the AuNSs had the same surface coating and comparable zeta potential as the AuNDs, the surface coating of the citrate-capped AuNSs was replaced with CTAB prior to their functionalization with the **C1** and **COOH** ligand mixture [44]. After surface modification with the **C1** and **COOH** ligands, their characteristics were analyzed by UV-Vis-NIR spectroscopy, DLS, TEM, and zeta potential (Appendix A). Zeta potential changed from highly negative (−41 mV) for citrate-coated AuNSs to highly positive (55 mV) for the CTAB-coated AuNSs, and to −9 mV for 40-AuNS-(100:0), which was very close to that for AuND-(100:0). Thermo-responsive assembly/disassembly was then investigated for 40-AuNS-(100:0). Upon heating, the LSPR peak of 40-AuNSs was markedly red-shifted between 40 and 50 °C (Figure 4a). This temperature range was higher than that for 105- and 60-AuND-(100:0). Upon cooling, the red-shifted plasmon peak returned to the original wavelength between 50 and 40 °C, indicating disassembly and redispersion of 40-AuNS-(100:0) and a little hysteresis (Figure 4b,c, Table 1). This is quite consistent with our previous research on AuNSs, even though the volume was much larger than that previously examined [37]. On the other hand, this result is in contrast with those for 105- and 60-AuND-(100:0), which utterly failed to disassemble upon cooling. Contrary to the thermo-responsive assembly of 105- and 60-AuND-(99:1), the introduction of 1% **COOH** ligand totally diminished the thermo-responsive assembly of 40-AuNS-(99:1) (Appendix A). From these results, we deduced that the flat surfaces of the AuNDs are critical for their unique thermo-responsive behaviors, not their size or volume. In other words, the thermo-responsive assembly of AuNDs is insensitive to their diameter, unlike that of AuNSs. Thus, one merit of AuNDs is that the plasmonic properties of AuNDs can be tuned by changing their sizes (diameter) without consequent changes in their thermo-responsive properties, a condition that cannot be realized with AuNSs [37,38].

### 3.4. Effects of COOH Content on the Thermo-Responsive Assembly of AuNDs

Next, the effect of **COOH** content on the thermo-responsive assembly/disassembly was investigated, as it is known that the incorporation of a hydrophilic residue, such as carboxylic acid, in the thermo-responsive polymers with a lower critical solution temperature causes an increase in responsive temperatures [52,53]. The 105-AuND-(99:1) showed similar assembly/disassembly behavior but a lower *T*_A_ compared to the AuND-(97:3) (Appendix A). The 105-AuND-(95:5) showed a higher *T*_A_ with smaller spectral changes in intensity compared to the AuND-(97:3) (Appendix A). A further increase in **COOH** content to 10% prevented thermo-responsive assembly; that is, 105-AuND-(90:10) did not show any spectral changes upon heating up to 70 °C (Appendix A). Likewise, the thermo-responsive assemblies of 60-AuND-(99:1), -(97:3), -(95:5), and -(90:10) were the same as that for 105-AuND (Figure 3 and Appendix A). Their *T*_A_ values were determined as the midpoint between the extinction changes during the heating process and plotted against **COOH** content and zeta potential (Figure 5a). It is clear that an increase in **COOH** content from 0 to 5% leads to an increase in assembly temperature from approximately 36 to 76 °C. The 60-AuNDs also showed similar results (Appendix A). This relationship between **COOH** content and assembly temperature suggests that 10% **COOH** could increase their assembly temperature to over 80 °C, which makes thermo-responsive assembly in water difficult.

Importantly, as noted above, a larger hysteresis on the thermo-responsive assembly/disassembly of AuNDs was observed (Figure 2c, Figure 3c, Appendix A). Thus, the assembly (*T*_A_) and disassembly (*T*_D_) temperatures of 105-AuNDs, 60-AuNDs, and 40-AuNSs are summarized in Table 1. The increase in **COOH** content leads to a larger degree of hysteresis based on the difference between *T*_A_ and *T*_D_, except for 105-AuND-(99:1) (Figure 5b). This could result from the suppressed increase in *T*_D_ against *T*_A_, probably due to the stabilization of their assembly via hydrogen bonding between carboxylic groups and/or carboxylic acid and the oxygen atom in the ethylene glycol units under the hydrophobic environment [41,54]. Further details on the hysteresis of the thermo-responsive assembly/disassembly are presented in the next section.

### 3.5. Mechanism for Hysteresis

To clarify the mechanism for hysteresis on AuND assembly/disassembly, that is, whether it is rate-dependent (kinetic) or independent (thermo-dynamic), we studied the effect of incubation time on the thermo-responsive assembly of 60-AuND-(97:3). Foremost, the temperature of the sample was kept at 25, 40, 45, 50, 55, and 60 °C and the time course of the extinction spectra change was measured every 30 min up to 90 min. The plot of the time course revealed no thermo-responsive effect when the sample was kept at 25 °C (Appendix A). A small decrease in peak intensity was subsequently observed up to 45 °C and significant decreases were observed at 50 °C. Notably, while the sample solution was at equilibrium at 60 min, the plot revealed a significant decrease in the extinction peak intensity (over 75% to be equilibrated) within 30 min, indicating that the first 30 min is critical for the thermo-responsive assembly of 60-AuNDs. As a sequel to the above result, a longer waiting time of 30 min was thought to be essential for the study of the hysteretic mechanism. Thus, we investigated the effect of time on hysteresis for 60-AuND-(97:3) assembly by waiting for 30 min during the heating and cooling of the sample between 25 and 70 °C. The results of the spectral analyses revealed a similar decrease/increase in the plasmon peak intensities upon the increase/decrease in temperature (Appendix A). Here, the 30 min waiting time was found to result in significant hysteresis (Figure 6a). A plot of the degree of hysteresis on the thermal cycling with a 5 and 30 min waiting time revealed some decrease, but the degree of hysteresis remained comparable for the longer waiting time (Figure 6b). We also studied the effects of a 5 and 30 min waiting time on the observed hysteresis for 60-AuND-(99:1) and similar results were also obtained (Appendix A). These results indicate that the hysteretic behavior of AuND assembly mostly results from a thermo-dynamic phenomenon [55]. Grzybowski et al. reported that hysteresis during the dispersion/aggregation of charged nanoparticles accompanying pH changes derive from a subtle interplay between electrostatic and van der Waals forces [56]. In our system, AuNDs could provide strong van der Waals forces and hydrophobic interactions between particles as indicated above. Hydrophobicity at the surface resulting from the dehydration of the HEG portion could inhibit rehydration due to the reduced water accessibility to the inner surfaces of the closely packed assemblies, but this could be a kinetic (rate-dependent) factor. On the other hand, as described in the previous section, the higher **COOH** content showed a higher degree of hysteresis. This is expected to be due to the increased formation of hydrogen bonding as a new interaction in the face-to-face assembly. The formed hydrogen bonding is stabilized under the hydrophobic environment from the dehydrated HEG molecules and could have contributed to the large hysteresis observed in this study. Thus, we attributed this hysteresis to Van der Waals interactions and hydrogen bonding.

In the light of the foregoing, it is suggested that the flat surfaces of AuNDs allow for the introduction of **COOH** ligands without loss of thermo-responsive assembly, leading to the above-demonstrated hysteresis. Conversely, the introduction of **COOH** ligands to the curved surfaces of AuNSs completely hinders their thermo-responsive assembly. The ability of AuNDs to retain their inherent features, even with the introduction of an electronegative moiety, is another merit of the flat plasmonic nanomaterial over AuNSs. These unique properties of AuNDs, among others, can be exploited to generate novel nanodevices for specific applications.

## 4. Conclusions

We successfully synthesized gold nanodiscs functionalized with different mixing ratios of **C1** and **COOH** ligands as HEG-derivatives. They showed a strong plasmonic peak at approximately 830 nm and significant changes in spectra in response to temperature, supporting the notion of thermo-responsive assembly via hydrophobic interactions due to the dehydration of the HEG portion in the ligands. STEM imaging and FDTD simulation support the presence of face-to-face structures on assembly. In contrast to the spherical nanoparticles modified with the same thermo-responsive ligand, AuNDs showed different assembly/disassembly behavior such as irreversible assembly for AuNDs modified with 100% **C1** ligand and a large hysteresis on reversible assembly/disassembly for AuND-(99:1), -(99:3), and -(99:5). In particular, AuNDs exhibited rate-independent hysteresis from different thermo-dynamic processes, including strong van der Waals interactions, hydrogen bonding in the hydrophobic environment, and so on. These results demonstrated the consequences of particle shape and interparticle interactions on the thermo-responsive assembly of isotropic and anisotropic plasmonic nanoparticles. This study demonstrated the unique thermo-responsive behavior of AuNDs and highlighted some potential functionalities that can be utilized, for instance, extracellular attachment of AuNDs for cell labeling and sensing as well as for cancer theranostic applications. Further, beyond bioapplications, AuNDs can be modified with diverse ligands to facilitate the design and fabrication of smart and functional nanodevices for advanced applications.

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
