# Peer review of "Hysteresis in the Thermo-Responsive Assembly of Hexa(ethylene glycol) Derivative-Modified Gold Nanodiscs as an Effect of Shape"

_nanomaterials, 2022, doi:10.3390/nano12091421_

Round 1

Reviewer 1 Report

This manuscript demonstrate the reversible assembly of gold nanodiscs using a thermo-responsive ligand. Overall, it is an interesting work, but there are many technical problems in the manuscript. It requires further improvement before publication.

  1. Please double check the accuracy of all images in the manuscript, especially the coordinate axis. I can spot a lot of mistakes in the figure. The figure caption should also be improved to provide more information and make the figure more readable.
  2. Please double check the accuracy of the abbreviations in the manuscript, such as “temperature (TD) temperatures” in Page 11.
  3. The concentration of ligand is a key factor to adjust the assembly process. A control experiment is expected for detail illustration.
  4. The mechanism also needs more experimental support. Other types of end-modified ligands can be employed to further elucidate the mechanism.
  5. What’s the experimental parameters for the assembly shown in Figure 1e and 1f? The SEM image in Figure 2 is contradictory to that in Figure 1 if the latter was dried at room temperature.
  6. The FDTD simulation on multi-layer AuNDs is expected.

Author Response

 We would like to thank the reviewers for their careful reviewing and valuable comments. To address these comments we have revised our manuscript. Please find our point-by-point responses and revisions according to the reviewer's comments shown in the file.

Reviewer 2 Report

This manuscript by Chidiebere et al. describes the preparation of surface-functionalised gold nanodiscs and investigations into their temperature dependences of plasmonic properties.  In my opinion, the reasons for the study are explained well, the methods are described in detail, experimental results are presented clearly and the conclusions are supported by those results.  I found no problems with the technical or scientific content.

There were only a few minor typographical errors, which I have listed below:

P4, 1st line: '...surface-modified AuNDs were washed by three...'

P5, 1st line: '...absorption spectra of the AuNDs in water...'

P7, L18: '...non-reversibility of the...'

P11, L9: '...and disassembly (TD) temperatures of...'

I expect these minor issues can be corrected at the proof stage.  Consequently, I am happy to recommend that the manuscript can be accepted for publication.

Author Response

We would like to thank the reviewers for their careful reviewing and valuable comments. To address these comments we have revised our manuscript. Please find our responses and revisions according to the reviewer's comments shown in the file.

Round 2

Reviewer 1 Report

It is a complete and convincing work. But the authors could have paid more attention to the details of writing the article.